# The Influence of Aminoalcohols on ZnO Films’ Structure

**DOI:** 10.3390/gels8080512

**Published:** 2022-08-17

**Authors:** Ewelina Nowak, Edyta Chłopocka, Mirosław Szybowicz, Alicja Stachowiak, Wojciech Koczorowski, Daria Piechowiak, Andrzej Miklaszewski

**Affiliations:** 1Institute of Materials Research and Quantum Engineering, Faculty of Materials Engineering and Technical Physics, Poznań University of Technology, Piotrowo 3, 60-965 Poznań, Poland; 2Institute of Physics, Faculty of Materials Engineering and Technical Physics, Poznań University of Technology, Piotrowo 3, 60-965 Poznań, Poland; 3Institute of Materials Engineering, Faculty of Materials Engineering and Technical Physics, Poznań University of Technology, Piotrowo 3, 60-965 Poznań, Poland

**Keywords:** sol-gel, ZnO, aminoalcohols, DEA, MEA, Raman, UV-Vis, photoluminescence

## Abstract

Preparing structures with the sol-gel method often requires control of the basal plane of crystallites, crystallite structures, or the appearance of the voids. One of the critical factors in the formation of a layer are additives, such as aminoalcohols, which increase the control of the sol formation reaction. Since aminoalcohols differ in boiling points and alkalinity, their selection may play a significant role in the dynamics of structure formation. The main aim of this work is to examine the properties of ZnO layers grown using different aminoalcohols at different concentration rates. The layers were grown on various substrates, which would provide additional information on the behavior of the layers on a specific substrate, and the mixture was annealed at a relatively low temperature (400 °C). The research was conducted using monoethanolamine (MEA) and diethanolamine (DEA). The aminoalcohols were added to the solutions in equal concentrations. The microscopic image of the structure and the size of the crystallites were determined using micrographs. X-ray diffractometry and Raman spectroscopy were used for structural studies, phase analysis and to establish the purity of the obtained films. UV-vis absorption and photoluminescence were used to evaluate structural defects. This paper shows the influence of the stabilizer on the morphology of samples and the influence of the morphology and structure on the optical properties. The above comparison may allow the preparation of ZnO samples for specific applications.

## 1. Introduction

Sol-gel synthesis for producing high-quality structures for electronics has gained massive popularity. The method provides the possibility of growing organic and inorganic materials with minimal equipment requirements. The process enables the creation of organic/inorganic hybrids, which may be useful as a protective coating [1], electrochromic devices [2], or solar cells [3]. One of the most frequently synthesized materials using the sol-gel method is polycrystalline and amorphous oxides such as zinc oxide (ZnO). ZnO polycrystalline materials play a considerable role in producing piezoelectric sensors, acoustic wave devices, UV LEDs, or as transparent conductive oxides [4]. To obtain the structure for specific requirements, it is crucial to control the orientation of the basal plane of crystallites, crystallite structures or the appearance of the voids [4]. Thus, understanding all the factors involved in a sol-gel synthesis that may influence the final product is necessary to prepare a highly scalable process. In the case of synthesis of the ZnO layers, those parameters may be dependent on the nature of the precursor and its concentration in the solution, the solvent used, and its acidity, or the method of layer deposition, substrate, thermal treatment, etc. [4].

The additives are critical factors in the formation of a layer and their homogeneity. Additives may act as an additional precursor solvent, facilitate the complex formation, stabilize the sol, or help in the formation of thin films [5]. Substances used as stabilizers often form ligands that attach to metal cations, either directly or by substituting other groups. Many anionic ligands include weak bonds with metal cations, which allow the initial structure to be determined by participating in the formation of bridges between metal atoms. In addition, using an anionic ligand enables the complexation process to be accelerated by displacing the water molecule [6] attached to the metal by the unstable addent.

One of the groups of substances that can participate in the formation of anionic ligands are the aminoalcohols, which, according to their alkali nature and bonding with the Zn (II) ion, allow control of the sol formation reaction [5,7]. In the case of sol-gel reactions, aminoalcohols tend to increase the solubility in the solvent, which is particularly important when selecting the solvent in terms of its boiling point or surface tension with substrate [4,8]. Aminoalcohols can act as amphoteric solvents [6]. They may create amphiphilic coatings around individual zinc acetate molecules, making the salt more soluble [9]. As a result of the chelating properties, aminoalcohols prevent the rapid precipitation of zinc hydroxide. The amine and hydroxyl groups coordinate the metal atoms in the alkoxides, which averts the uncontrolled process of hydrolysis [4,8,10,11]. Amino alcohols may act as two ligands, which allows the coordination of zinc atoms through a dual action—chelating and bridging the two atoms [4].

The presence of amines increases the alkalinity of the medium. Control of the pH of the solution by adding aminoalcohols allows for a particular prediction of the layer’s behavior. Addonizio et al. noticed that more alkaline solutions will enable one to obtain a more porous structure, with higher thickness and crystallinity than in the case of films grown from a more acidic solution [12]. Thongsuriwong et al. noted that for the growth of nanostructures, the higher pH achieved by the addition of MEA and DEA makes the structures larger and more homogeneous compared to particles grown in solutions with lower amine concentrations [10].

The purpose of this paper is to examine the properties of ZnO layers grown using the most commonly used stabilizers and aminoalcohols in various concentrations. For a better comparison, the layers were grown on different substrates, which would present additional information about the behavior of the layers on a specific substrate.

The research was conducted using monoethanolamine (MEA) and diethanolamine (DEA) in equal percentage concentrations. MEA (H${}_{2}$NCH${}_{2}$CH${}_{2}$OH) has in its structure an ethanol molecule in which one of the hydrogens is replaced with an amino group. The boiling point of the MEA is about 170 °C. When the molecule acts as a bridge between zinc atoms; the high boiling point prevents its degradation during structure dehydration or solvent evaporation. It also allows for maintaining an equal crystallization temperature [13], or controlled oxide reduction through the aging process [14]. Khodja et al. observed a decrease in grains on the sample surface and a change in the optical band gap with an increase in the concentration of MEA relative to the precursor [15]. These observations were confirmed by the studies of Yahia et al., where with the decrease in the precursor concentration, the value of the energy gap [16] decreases.

DEA ((CH${}_{2}$CH${}_{2}$OH)${}_{2}$NH) consists of two ethanol molecules linked together by the amine bridge. DEA has a high boiling point (about 269 °C [17]), which means that the layers made with its use must be heated at a high enough temperature to be able to remove any DEA decay products from the layer [11]. It has been suggested that in reactions using DEA as a stabilizer, Zn (II) and diethanolamine form complexes due to ligand exchange between Zn(OAc)${}_{2}$ and H${}_{2}$ [18]. As with MEA, a higher addition of a stabilizer produces layers with higher crystallinity [11]. According to the literature [8,14], this stabilizer has a noticeable effect on the layer—it is possible to achieve much higher crystallinity by using it.

In the production of layers with aminoalcohols, the complete decomposition of stabilizing substances has a crucial role, as their presence degrades the optical and electrical properties [7]. Therefore, the purpose of the article is to examine the possibility of using individual aminoalcohols in order to obtain the desired properties and specific purity of the ZnO layer. The paper presents the impact on the sample’s morphology and the influence of the morphology and structure of the sample on the optical properties. It also claims problems that may arise from the use of specific stabilizers, such as the formation of additional phases or the lack of complete degradation of the by-products, which can have an impact on the structural, optical, and electrical properties of the films.

## 2. Materials and Methods

### 2.1. Preparation of Samples

Samples were grown on three different substrates (n-type (001) silicon (Si), (0001) sapphire (Al${}_{2}$O${}_{3}$), and quartz glass (assigned as SiO${}_{2}$) obtained from Shanghai Famous Trade Co. LTD. The layer fabrication method is a modified method described already by Nowak et al. [19]. The preparation of a precursor solution began with dissolving zinc diacetate dihydrate (Zn(C${}_{2}$H${}_{3}$O${}_{2}$)${}_{2}\cdot$2H${}_{2}$O) in 10% and 5% solutions of MEA/DEA in 2-methoxyethan-1-ol (C${}_{3}$H${}_{8}$O${}_{2}$) to prepare a 1M solution of zinc diacetate. The prepared suspension was stirred for 1 h and aged at 50 ° C for 24 h.

All substrates were cleaned in four steps: in the solution of detergent and ultrapure water obtained with a Milli-Q water purification system (electrical resistivity 18.2 MΩ·cm), methanol, with the RCA-1 bath, which is described in more detail in [20], and the 5% piranha solution in water to remove the oxidized layer. After the cleaning, the substrates were rinsed in ultrapure water and acetone and dried in hot air.

The films were deposited using the spin coating method. After setting the substrate on the spin coater’s disc, the coating solution (0.1 mL) was dropped on the substrate and spin-coated in three steps at 500 rpm (4 s), 1500 rpm (5 s), and 3500 rpm (25 s). Such centrifugation parameters were determined to be sufficient to achieve uniform coverage of all substrates by all prepated solutions. After coating, the film was dried at 100 °C in a hot oven for 10 min. The process was repeated eight times for each sample. The prepared films were annealed at 400 °C in a tube furnace for 1 h and left in the furnace until cooled to crystallize without superficial fissures and points. The specific temperature was chosen to eliminate the influence of subphase between the layer and the substrate which can happen at a higher heating temperature [19]. What is more, the temperature was sufficient for degradation and evaporation of both stabilizers (the boiling points of additives are 271 °C for DEA and 171.1 °C for MEA) [4]. The one-hour annealing period was the shortest time to achieve polycrystalline layers from sol. A detailed description of the preparation of each sample is included in Figure 1.

### 2.2. Instrumental Analysis

The crystallographic structure estimation and phase analysis of ZnO samples were obtained with X-ray diffraction (XRD) conducted using the Panalytical Empyrean equipment with a copper anode (CuKα, 1.54 Å) at a Bragg–Brentano reflection mode configuration with 45 kV and 40 mA parameters. The measurement parameters were set up for 20–90° with 15 s per step and 0.0167° in all cases. Due to its high sensitivity to changes in crystal structure and orientation, Raman microscopy measurements were conducted for a more detailed study. Raman scattering was observed at room temperature using the Renishaw inVia system. The spectrometer was equipped with Leica confocal mA long-distance objective with 50 times magnification, and an Argon laser at 488 nm as the excitation light source. The Leica microscope with the same objective was also used to assess the microscopic appearance of the layer. The distribution and mean size of crystallites in samples grown on Al${}_{2}$O${}_{3}$ and Si were estimated from the micrographs obtained with the Scanning Electron Microscope (SEM). The SEM images of each sample were recorded using the FEI Helios NanoLab 660 tool at 10 kV electron beam energy. The grain size was calculated using the ImageJ software by measuring for lengths of 1000 randomly selected microcrystals in each sample. The crystallite mean sizes were calculated with arithmetic mean and standard deviation for error evaluation. Finally, the thickness of the films was estimated with a Zeiss LSM 710 confocal microscope by measuring the distance between focused planes of substrate and layers.

A Varian Cary 4000 with a halogen and deuterium lamps was used for the absorption measurements. Photoluminescence spectra were obtained using the Hitachi F-4500 system equipped with a xenon lamp as an excitation light source with the excitation wavelength at 330 nm. All measurements were performed at room temperature.

## 3. Results

### 3.1. Layer Morphology and Structural Properties

The optical and electrical properties of polycrystalline layers depend not only on structural defects or impurities but also on the size and shape of the grains. Due to the shape, the grain exhibits different amounts of so-called “dangling bonds” at the boundary of a single grain [21]. Particle size is usually controlled by the recrystallization of the layers at a specific temperature. However, the size of individual crystallites also varies depending on the stabilizer used. Therefore, microscopic images were analysed to assess the effect of stabilizers on the microscopic image of ZnO layers, films’ thickness, and grain size distribution.

The Figure 2 shows the layers deposited on silicon using a solution based on DEA (concentration 10% (A) and 5% (B)) and MEA (concentration 10% (C) and 5% (D)). As can be observed, the samples made with the use of DEA are characterized by a much smaller grain than the samples made with MEA stabilized solutions. Moreover, these layers are much more homogeneous than the samples prepared with the MEA solution. The concentration of the stabilizer strongly influences the granularity; a sample made from a solution with a lower concentration is characterized by a larger grain with more distinct boundaries between the crystals.

Figure 3 exhibits the layers embedded in the sapphire. All films have a much smaller crystallite than the layers deposited on silicon. Moreover, the layers made with MEA have much larger grains than those synthesized with DEA. As in the case of silicon-deposited layers, structures with a lower concentration of stabilizers have a more pronounced and more prominent grain. In the case of samples deposited on quartz (Figure 4), the structure deposited from DEA solution has a much greater homogeneity than the sample produced with MEA. This sample is characterized by a much smaller granularity on the surface. As may be concluded, the samples stabilized with MEA are characterized by large, distinct crystallites and high surface heterogeneity. The deposition layers of the DEA solution used are homogeneously covered with a layer of smaller grains with an indistinct grain boundary.

The SEM micrographs for samples grown on Si and Al${}_{2}$O${}_{3}$ are shown in Figure 5 and Figure 6. As may be observed, the crystallites of the MEA samples create local densities that resemble folds. Comparing the structures formed by DEA, they are slightly more homogeneous on the analyzed surface. The distribution and mean size of crystallites differ in terms of the stabilizer used but also in the substrate. In the case of samples grown on the sapphire, the sample grown with a 5% concentration of MEA exhibits the biggest shift towards small lengths. Furthermore, in the distribution of samples grown from solutions with a 5% concentration of MEA and a 10% concentration of DEA, crystallites of 50–57 nm are predominant. For samples fabricated from a solution with 5% DEA, crystallites of 75–100 nm predominate, while the distribution of the 10% MEA sample is dominated by microcrystallites of 100–150 nm.

The mean size of crystallites is shown in Table 1. As may be observed, the size of the crystallites has no direct relation to the concentration or the type of stabilizer. It is worth noting that the samples based on MEA form local groups of crystallites, which results in the appearance of large grains, which were observed under an optical microscope. Thus, it can be said that in the case of MEA, the crystallites tend to create larger fractal structures, while in the case of DEA, the structures remain more or less homogeneous.

The intensity of spectral features strongly depends on the thickness of the sample. Thus, for all layers, the rough thickness was determined by measuring the change in focus height using confocal microscopy. The individual thicknesses are shown in Table 2.

The influence of annealing time on the ZnO structure is shown in an XRD pattern (Figure 7). The analyzed films exhibit the wurtzite (P6${}_{3}$mc in Hermann–Mauguin notation) symmetry and a polycrystalline character. For most samples, the phases correlated with the intermediate products are present, such as various phases of Zn(OH)${}_{2}$ [22,23] or nitrogen compounds such as H${}_{4}$N${}_{2}$O${}_{3}$ [24] or NH${}_{4}$N${}_{3}$ [25]. Interestingly, samples produced with DEA exhibit a very high content of Zn(OH)${}_{2}$ phase-ZnO related peaks in the case of samples grown from a solution with 10% DEA content, and are very weak compared to the peaks related to the intermediate phase. Due to the high boiling point of DEA, it is very possible that DEA-based structures require a much longer aging time compared to samples based on MEA. In the case of MEA samples, as may be observed, the prominent ZnO peaks are observed, with the (002) as a preferred orientation.

The lattice constants were calculated from the peaks from wurtzite ZnO phase. The constant *a* was estimated from the peaks with hk0 and constant *c*-for bands where indicators h, k = 0. In case of constant a, the standard deviation determined the measurement error, but in the case of constant *c*, the error was calculated using the logarithmic differentiation method. The lattice parameters are presented in Table 3. The lattice constant *c* increases strongly for samples grown on silicon with DEA. What is more, the c/a ratio increases both, with the increase in stabilizer concentration and for DEA samples. A higher *c*/*a* ratio indicates a lower compression of the atoms towards the *c* axis and lower *c*/*a* values indicate a higher packing of atoms concerning the c-plane. The elongation of the c-axis was already observed by Ben Aziza et al. [26] in the case of samples grown with the stabilizer with a higher content of nitrogen and was assigned to microstrains caused by nitrogen bonding to ZnO lattice. Interestingly, the *c*/*a* ratio is higher for samples grown on silicon, which can be caused by a much larger lattice mismatch compared to sapphire.

### 3.2. Raman Analysis of Layers on Different Substrates

Raman microscopy studies were performed for all layers to analyze changes in the surface caused by layer deposition. The most stable form of ZnO is the wurtzite structure (with lattice parameters a = 3.24 Å and c = 5.21 Å) [27]. Room-temperature Raman scattering spectra demonstrated transverse optical phonons (TO) at 378 cm${}^{-1}$ (A${}_{1}$) and 410 cm${}^{-1}$ (E${}_{1}$), longitudinal optical (LO) modes at 576 cm${}^{-1}$ (A${}_{1}$) and 588 cm${}^{-1}$ (E${}_{1}$), and E${}_{2}$ phonons at 98 cm${}^{-1}$ (low), and 438 cm${}^{-1}$ (high) [28]. Depending on the Raman selection rules, the Raman modes observed in the spectrum consider the crystal orientation relative to the directions and polarization of the incoming and outgoing light [29]. The samples were analyzed on their growth surface. Because of the low thickness, the measurement enabled the registration of all substrates and ZnO- related peaks. The intensity of the signal depends on the thickness of the layer and the size of the crystallites; therefore, the intensity of the signal varies with the additive used.

Figure 8 shows the spectra for the layers produced on sapphire. Due to the substrate’s nature, all spectra made for the layers deposited on the sapphire are characterized by very strong bands coming from vibrations in the Al${}_{2}$O${}_{3}$ crystal lattice (bands marked in orange). In the spectra, three bands from the ZnO layer can be observed, which are in turn the E${}_{2}$ (high)-E${}_{2}$ (low) band located at 328 cm${}^{-1}$, the E${}_{2}$ (high) located at 436 cm${}^{-1}$, and the second harmonics of LO modes located in the region of 1030–1200 cm${}^{-1}$. As can be observed, the MEA stabilized layers are characterized by a much higher intensity of the ZnO modes concerning the sapphire bands than the DEA stabilized layers. When DEA is used, the only vibration mode observed in the ZnO lattice is the weak signal from the E${}_{2}$ (high) band, which is the most intense band for the Z (XX) Z geometry.

In the case of the spectra obtained for the samples produced on silicon, due to the very strong signal received from the substrate (Figure 9A), it was necessary to limit the analysis only to the range of the prominent band appearance for the ZnO layer (Figure 9B) (E${}_{2}$(high) located at 436 cm${}^{-1}$), and to improve comparability, all spectra were normalized to one concerning the most intense band for silicon (located at 520 cm${}^{-1}$). As in the case of the layer grown on Al${}_{2}$O${}_{3}$, the structures from the MEA stabilized solution demonstrate much higher intensities of the ZnO-derived band compared to the layers grown from DEA-based solutions, for which the intensity of these bands are only slightly above the noise level.

Due to the low relative intensity of the ZnO peak with respect to the substrate bands, the ratio of the most prominent ZnO peak (E${}_{2}$(high)) and the most intensive modes of the crystal substrate was juxtaposed in Figure 10. As may be observed, the ZnO/Si peak intensity ratio is much less intensive than ZnO/Al${}_{2}$O${}_{3}$. Nevertheless, it is worth noticing that in both cases, the samples grown with MEA stabilizers exhibit more intensive signal than DEA ones. The much weaker ZnO spectrum in the case of all layers deposited with DEA on crystal substrates may be affected by three factors. Firstly, all of the DEA samples have lower thickness than the MEA ones, so the signal from the film may be much weaker than the signal from bulk substrates. Moreover, all DEA samples exhibit additional phases, which may weaken the signal intensity. Lastly, the weakening of the signal may be correlated with the destruction of long-range order through defects or local damage to the layer. According to the literature, this can both lead to weakening of the intensity of the E${}_{2}$ phonon peak and lead to broadening of the spectral line [30].

Figure 11 presents the spectra for the quartz-deposited layers. Quartz is an amorphous structure that has relatively weak and broad Raman bands. All bands related to ZnO vibrations are marked. The peak for the E${}_{2}$ (high) mode is the most intense band, located at 437 cm${}^{-1}$. The spectrum shows the peaks from the differential mode E${}_{2}$(high)-E${}_{2}$(low) located at 330 cm${}^{-1}$, bands of low intensity in the area related to the LO components of the polar modes (and their second harmonics) or the second harmonic of the B${}_{1}$(low) mode, which is activated by N atoms embedded in ZnO lattice [31]. The B${}_{1}$(low) mode signal is more prominent for DEA sample in comparison to MEA layer, which indicates a much more significant defect in the layer produced with the use of DEA. At the same time, it is worth noticing that peaks at 1400 cm${}^{-1}$ (rocking and scissor vibrations of CH${}_{2}$ [32] bonds) and 2900 cm${}^{-1}$ (stretching vibration of CH${}_{2}$ [32] bonds) appeared in spectra obtained from a layer made with DEA. Those bands indicate a lower degree of degradation of organic substances under the influence of annealing. This feature can be directly correlated with the much higher boiling point of DEA than MEA, making it necessary to anneal at higher temperatures to obtain a layer free of organic substances. Figure 12 shows the shifting of the E${}_{2}$(high) mode with respect to the stabilizer and substrate used. In general, the frequency shift may indicate the purity of the material, as when the packing of the molecules in the crystal or atomic masses changes, the intermolecular distance changes. Thus, the intermolecular force contents change, resulting in a frequency shift.

As may be observed, the E${}_{2}$(high) mode has a higher Raman shift value for samples grown on sapphire and quartz substrates than on silicon. The E${}_{2}$(high) mode of samples grown on silicon is wider, which can be correlated to the layer lower ordering of the atoms and their pursuit of the amorphous phase. A lower frequency shift may evidence higher interatomic spaces in comparison to samples grown on sapphire and quartz. Interestingly, samples grown on Al${}_{2}$O${}_{3}$ and SiO${}_{2}$ exhibit almost linear change in the E${}_{2}$(high) Raman shift in case the stabilizer and their concentration. Samples grown with DEA solutions are significantly blue-shifted compared to samples grown with MEA, which is consistent with the higher c/a values of samples grown with DEA. Similarly, lower concentrated solutions resulted in less elongated c-constant and Raman frequency blue-shifting. The observation excludes the effect of changes in the lattice packing. However, the shift may be affected by the incorporation of a smaller mass impurity in the network of atoms in the case of samples grown on the basis of DEA or by the usage of a higher concentration of stabilizers such as hydrogen or nitrogen.

### 3.3. Change in Spectral Properties Depending on the Stabilizer

The specific combination of Zn and O in crystallites or defects generated in a structure leads to the creation of additional energetic states [33]. Due to the structure of the grain atomic boundaries, crystallites may vary in a dislocation type, deformation of the crystal lattice, or dangling bonds [21,34]. The changes in electrical characteristics are attributed to the accumulation of the unsaturated charge of the dangling bonds [21]. Due to different pH levels, boiling points, or the mechanism of creating bridge bonds, the chemical composition of the precursor solution may have a substantial impact on the crystallization of the film [4]. Thus, UV-Vis absorption and photoluminescence were used to examine additional states created by the defects and disorders in the lattice.

Figure 13 presents the UV-Vis absorption and photoluminescence spectra for samples grown on sapphire. As shown in Figure 13A, the samples are characterized by a different degree of transparency, which depends on the stabilizer. The layers made using MEA have higher absorption than the samples made using DEA. Furthermore, the sample produced at a lower concentration of MEA has the highest absorption in the ultraviolet (UV) range. The gentle descent of the peak indicates that a wide visible range is covered. Moreover, samples produced with DEA exhibit a shift of the maximum of the most intense peak of the spectrum towards higher energies. The photoluminescence spectra are characterized by a broad band in the violet-UV region and additional bands in the visible range. The spectra differ significantly in intensity. Samples made with MEA have the highest luminescence intensity. For these layers, the band originating from the near-band edge emission is an extended band with a visible overlap of three curves with a maximum of 3.16 eV, 3.07 eV, and 2.95 eV. Bands with lower energies may be associated with transitions from the acceptor level of interstitial zinc (Zn${}_{i}$) to the state associated with the complexes V${}_{Zn}$− V${}_{O}$ [35,36]. The samples exhibit broad bands in the visible range with estimated peak maxima at 2.75–2.65 eV and 2.35 and 2.04 eV. The peaks at 2.75–2.65 eV may be directly related to the oxygen vacancies. Regardless of the heat treatment process in an oxygen-rich atmosphere, due to short-term heating at low temperatures, the samples may not have been fully oxidized. The band in the 2.35 eV range is related to the appearance of zinc vacancies inside the sample [28,36]. In contrast, the luminance around 2 eV was assigned by Ke et al. [36] to the surface effect—the transition between the V${}_{Zn}$ and OH states, considering that OH groups adsorbed on the surface of a sample.

The samples based on the DEA solution are characterized by a much less intense glow and shifting the NBE-related band maximum by as much as 0.1 eV towards higher energies, which may indicate a broadening of the optical band gap.

The dependence of the spectral properties on the stabilizer used in the solution for the samples deposited on quartz looks similar to the samples made on Al${}_{2}$O${}_{3}$. As shown in Figure 14B, the absorbance spectra of the sample made from the MEA solution is higher in the visible range compared to the sample based on DEA. Moreover, the most characteristic absorption band is significantly broadened for MEA, and its estimated maximum is redshifted from DEA. The MEA-based sample has a much higher intensity of photoluminescence in the visible range than the DEA ones. The NBE band maximum for the DEA sample is slightly shifted towards higher energies, confirming earlier observations in which DEA-based samples demonstrated somewhat higher optical band gap values. Samples have broad luminescence bands in the visible range. Layers exhibit bands with maxima which could roughly be assigned to 2.65, 2.5, 2.3, and 2.0 eV. These features are consistent with the observations for the sapphire-grown samples.

Regarding the non-transparent substrate, in the case of samples grown on silicon, the analysis was conducted only for photoluminescence measurements. The results are shown in Figure 15. As can be observed, the samples grown from DEA-stabilized solutions exhibit very low luminescence, with a single, very weak peak with a maximum at 3.29 eV. The band for the MEA-based samples is, in turn, strongly redshifted. Moreover, the influence of MEA concentration on the spectral properties of the sample can be noticed: the reduction of the stabilizer concentration resulted in a shift of the main band from 3.25 eV (1 MEA) to 3.14 (0.5 MEA) and a broadening of the band. According to the previous predictions, the features may indicate additional transitions, which can be assigned to transitions on Zn${}_{i}$ [35,36]. Both samples show a completely different character in the visible range. The sample grown from a solution with a lower concentration of MEA has one strong band with a maximum at 2.34 eV, while sample 1 MEA has three strong bands at 2.65 eV, 2.5 eV, and 2.34 eV. The band at 2.34 eV can be assigned to zinc vacancies inside the sample [28]. The bands at 2.65 and 2.5 were, in turn, assigned to the generation of oxygen vacancies [28,36].

## 4. Discussion

In recent years, the influence of stabilizers on the ZnO structure has been highly investigated. However, the reaction dynamics strongly depend on the growth conditions. Thus, in the case of the experiment described, the structural changes in the samples, which were synthesized from a solution with a high solvent content, were investigated. The layers were annealed for 1 h in a furnace with a stable temperature of 400 °C and gradually chilled in a furnace until they were fully cooled down. The emergence of similar studies allows us to compare the changes in the case of samples deposited on different substrates and with other solutions. Furthermore, the comparison may allow for a realistic determination of the influence on the samples’ structure, or optical properties of the samples.

At first, the examination of the ZnO layers’ micrographs of produced samples was carried out. The general observations of the grain size correlate with the influence of concentration and the stabilizer used. The higher concentration of aminoalcohol leads to a smaller grain size. However, the observation made by previous research is quite vague. Verma et al. [11] observed the grain size increase with the higher DEA concentration.

On the other hand, in the case of Sivakumar et al. [37] research, the higher pH (which can be obtained by increasing the concentration of aminoalcohol) supported the growth of crystallites on ZnO films. As a result, the samples grown in alkali conditions had larger grains and were more homogeneous. The observation was confirmed by Boudjouan et al. [38]. They observed a smaller grain size for a lower concentration of MEA in the solution and a lower tendency to agglomerate to a few larger crystallites.

In the case of analyzed samples, the two characteristics were noted. First, the micrometric granular structure of the samples was more homogeneous in the case of samples grown with DEA solutions. In the case of nanometric changes, the concentration and type of stabilizer did not have a significant effect on the size of individual crystallites. However, MEA samples demonstrated a tendency to form a local compaction of the grains, which resulted in the appearance of a very granular micrometric structure. The stabilizer and its concentration have a significant influence on the thickness of the ZnO layer. The XRD patterns show the stabilizer’s strong influence on the formation of ZnO. MEA samples exhibit peaks correlated with the ZnO phase (Ref. code: ICDD:04-008-8199). However, samples grown with DEA show very prominent peaks derived from other phases such as Zn(OH)${}_{2}$, which is an intermediate in sol-gel synthesis. The presence of this phase can significantly influence the signal obtained from spectroscopic measurements. Disordering by the presence of additional atoms and incorporation into the structure of the additional phase influenced the elongation of the constant *c*, which also proved the change in the structural phase of ZnO. One of the possible dopants may be nitrogen (N), which may bond with Zn. The annealing process leads to elongation of the Zn-N bond (whereas the Zn-O bond length does not change) [39]. It may lead to microstrains, which, in the case of DEA grown samples in the work of Ben Aziza et al. [26], caused the prolongation of the c-axis. Microstrains are caused by tensile stress induced by the electrostatic interactions between Zn and O ions. These interactions are strongly correlated with the distance between ions, which is associated with changes due to the appearance and elongation of Zn-N bonds. The observation of a much smaller grain in the case of DEA samples and its origin can be confirmed with the Raman measurements. All DEA samples exhibited much lower Raman intensity compared to MEA ones. It can be correlated with poorer long-range ordering due to defects or layer degradation. As was previously estimated [9,40], DEA samples annealed even in higher temperatures exhibit a noticeable content of N, which can influence the long-range order in crystallites. On the other hand, the lower intensity may be correlated to the higher boiling point of DEA. DEA solution has a lower tendency to degradate at lower temperatures [13], which can be an advantage in the controlled reduction of oxides in a temperature [41]. On the other hand, the lower intensity and red-shifting of ZnO E${}_{2}$(high) mode in the case of DEA samples may be also connected with the appearance of the hydroxide phase in the structure, which was observed in XRD patterns.

The measurements for the samples deposited on the quartz follow the observation of Gomez-Nunez [9,40]—in the case of samples produced with the use of both stabilizers, hydrocarbon bands were recorded on the Raman spectra. However, the intensity of those peaks was much higher in the case of the DEA sample, which may indicate incomplete degradation under the influence of temperature. In the case of Gomez-Nunez measurements, the DEA stabilized sample was annealed at 600 °C, and it still demonstrated CH${}_{2}$ and CH${}_{3}$ bending vibrations in the IR spectrum.

In the case of UV-vis spectroscopy, the influence of substrate is visible—the maxima of NBE emission peaks shift from 3.16 eV for samples on sapphire substrates to 3.2 eV for quartz and vary from 3.14 to 3.29 eV in the case of silicon samples. According to literature, a higher concentration of aminoalcohols should lead to higher E${}_{g}$ values [42] and lower transmittance [43]. In the case of our measurements, changes influenced by precursor concentration are unclear and differ depending on the substrate. Therefore, this analysis was omitted in these studies. The more defined change occurs concerning the stabilizer used. In the case of samples grown on all substrates, the blue-shift of absorption edge for DEA samples may be registered, which was observed previously [44]. It is worth noting that this observation, considering the literature reports, is not repetitive. Ben Aziza et al. [26] observed the opposite trend for samples based on specific amino alcohols. All samples exhibit luminescence peaks originating from defects such as zinc interstitial and zinc and oxygen vacancies. Thus, it may be concluded that the annealing conditions induce these defects and are directly correlated with the stabilizer and annealing atmosphere.

## 5. Conclusions

This work presents a detailed analysis of ZnO structures stabilized with different aminoalcohols. This research aimed to compare the structural and optical properties of ZnO samples prepared with the sol-gel method, which were deposited on various substrates and annealed under different conditions.

Because the properties of polycrystalline films depend not only on structural defects or impurities but also on the grains’ size, the analysis focused on the layer micrographs of samples prepared on each substrate. The stabilizer used did not demonstrate any influence on crystalline size, but MEA solutions tend to create local compaction of the grains. The concentration and stabilizer had a significant influence on the thickness of the ZnO films and the stoichiometric composition.

The MEA-based structures exhibited more prominent Raman spectra. These features were associated with the elongation of Zn-N bonds and microstrains; incomplete degradation of DEA could have contributed to the deterioration of the layers’ properties, a change in their crystal constants, or the appearance of defects or even appearance of CH${}_{2}$ groups in Raman spectra or the presence of an intermediate phase of the sol-gel process.

The change in the lattice parameters correlated with the additional phases may be related primarily to the blue-shifting of the absorption maximum and the NBE band in the photoluminescence spectra. The feature is clear in the spectra comprehension for different stabilizers as well as in different concentrations. Regardless of the stabilizer, all samples exhibited PL bands related to transitions in the Zn${}_{i}$, V${}_{o}$, and V${}_{Zn}$ states. Thus, the appearance of those defects may be directly correlated with the annealing condition and the composition of the precursor solution.

The MEA stabilized samples present more prominent and grainy structures, which may become free from unintentional doping, impurities, and contamination during the annealing process in lower temperatures (<400 °C). DEA provides more homogeneous structures. Unfortunately, due to the high boiling point, these structures require temperatures higher than 400 °C for complete degradation and completion of oxidation of the hydroxy bridges.

## Figures and Tables

**Figure 1 gels-08-00512-f001:**
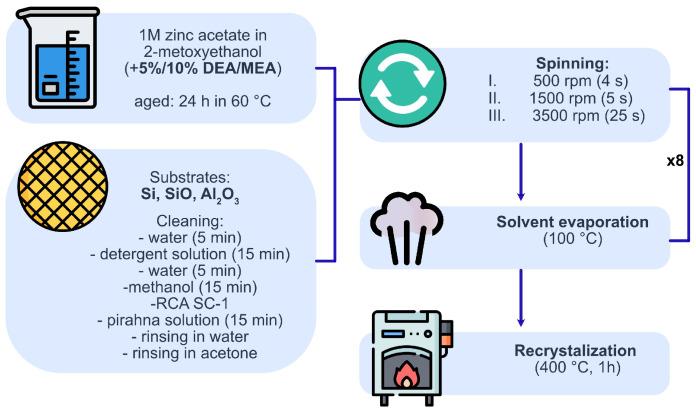
The algorithm of samples’ preparation.

**Figure 2 gels-08-00512-f002:**
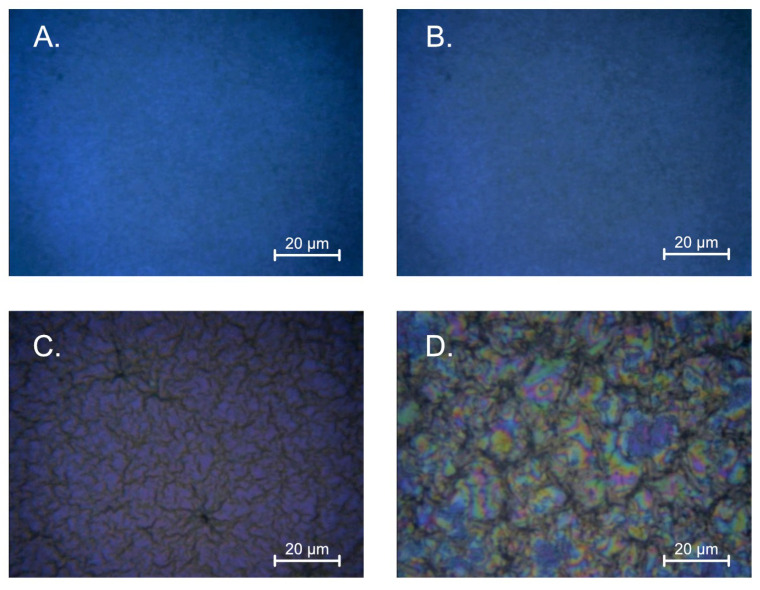
Summary of the morphology for the layers produced on silicon (**A**) using a solution with 5% concentration of DEA; (**B**) using a solution with 10% concentration of DEA; (**C**) using a solution with 5% concentration of MEA; (**D**) using a solution with 10% concentration of MEA.

**Figure 3 gels-08-00512-f003:**
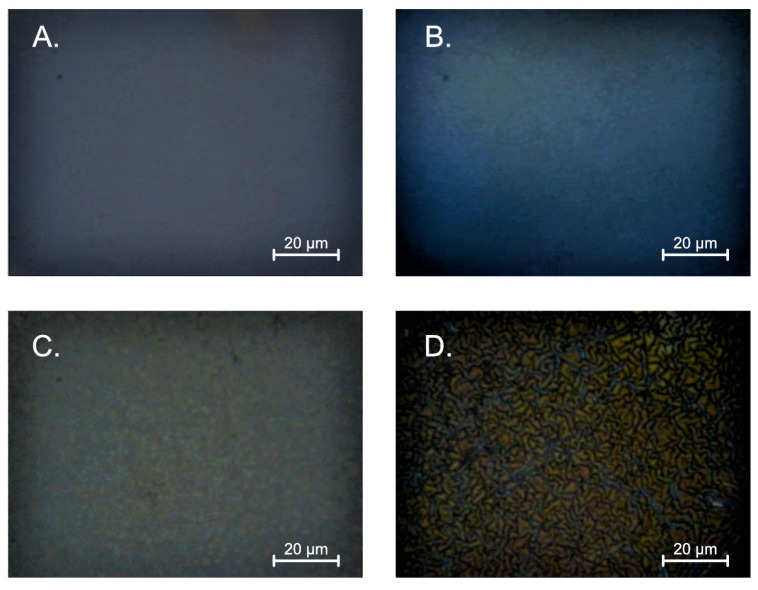
Summary of the morphology for the layers produced on sapphire (**A**) using a solution with 5% concentration of DEA; (**B**) using a solution with 10% concentration of DEA; (**C**) using a solution with 5% concentration of MEA; (**D**) using a solution with 10% concentration of MEA.

**Figure 4 gels-08-00512-f004:**
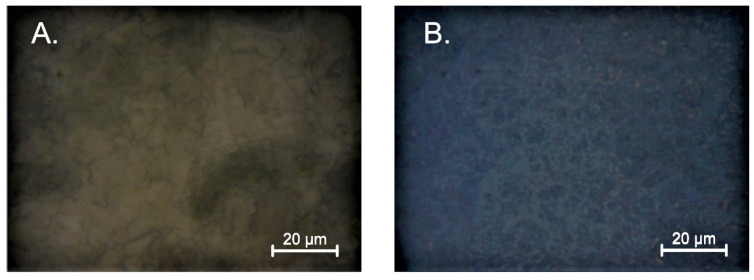
Summary of the morphology for the layers produced on quartz (**A**) using a solution with 5% concentration of DEA; (**B**) using a solution with 5% concentration of MEA.

**Figure 5 gels-08-00512-f005:**
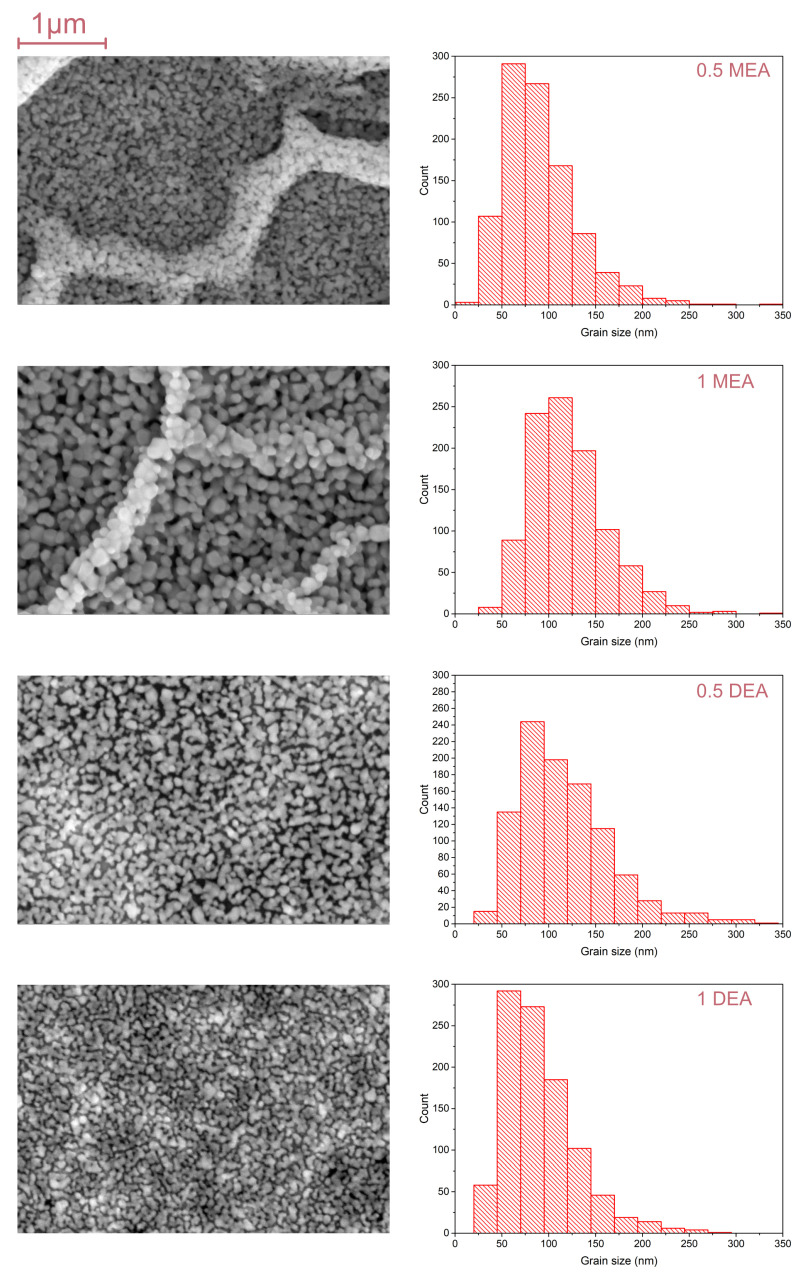
The SEM images and grain size distribution for ZnO layers grown on sapphire.

**Figure 6 gels-08-00512-f006:**
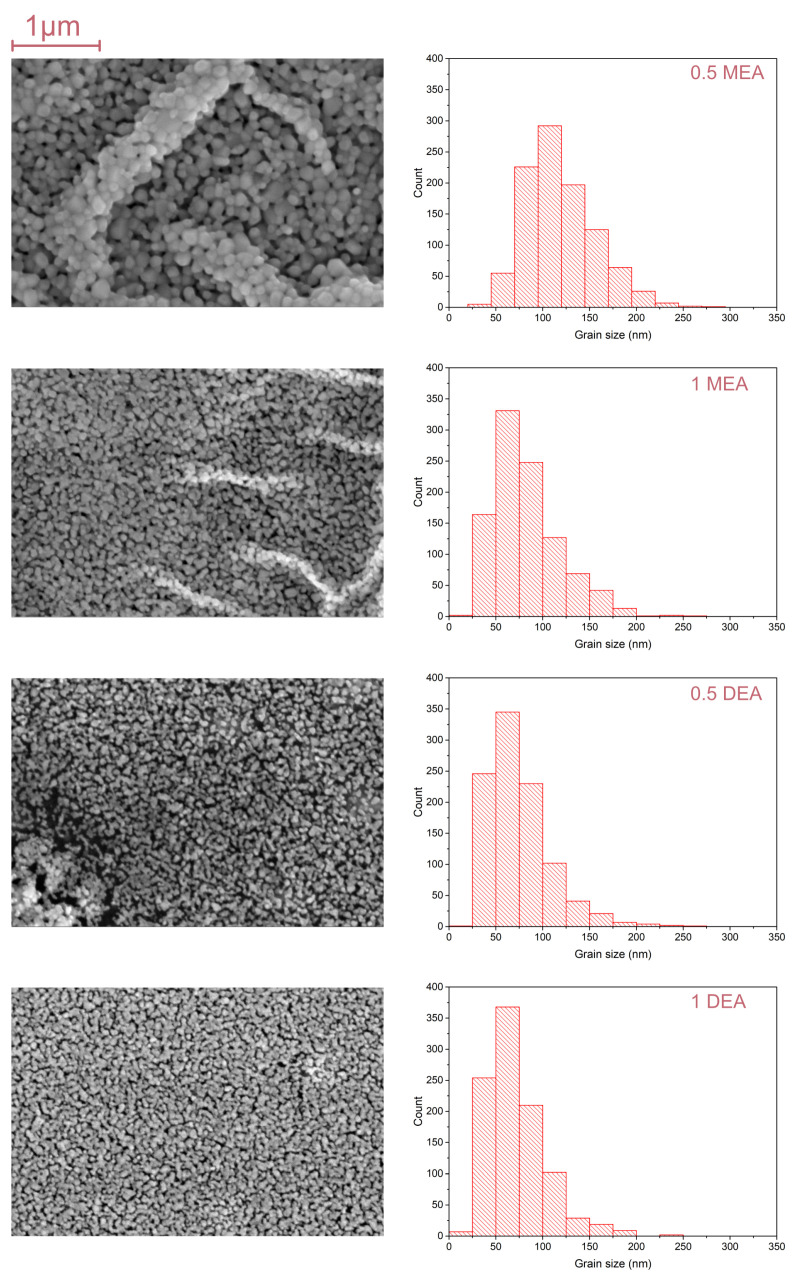
The SEM images and grain size distribution for ZnO layers grown on silicon.

**Figure 7 gels-08-00512-f007:**
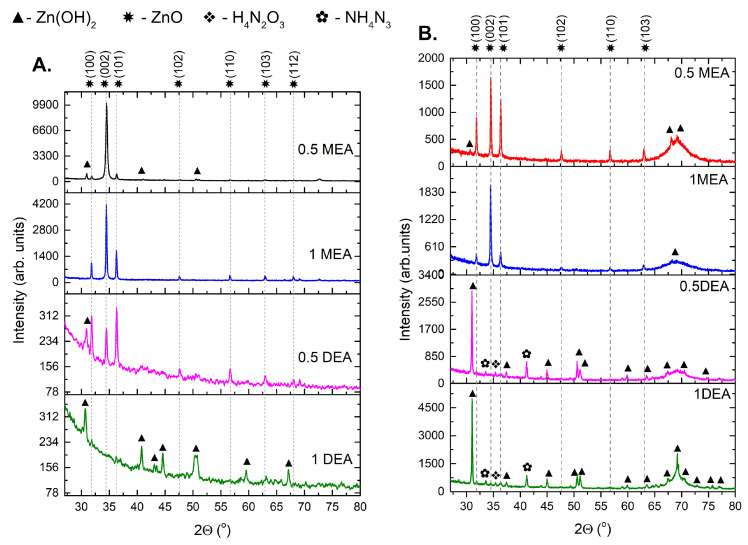
XRD pattern for samples grown on: (**A**) sapphire and (**B**) silicon. The gray lines relate to the position of the ZnO peaks; the rest of the phases are described with symbols.

**Figure 8 gels-08-00512-f008:**
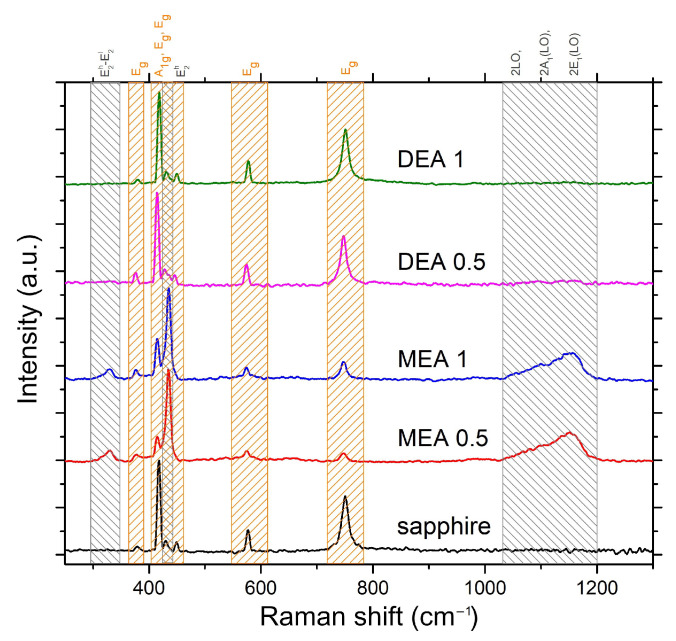
Raman spectra for samples grown on sapphire. The grey tags relate to the ZnO bands and orange-to the Al${}_{2}$O${}_{3}$ peaks.

**Figure 9 gels-08-00512-f009:**
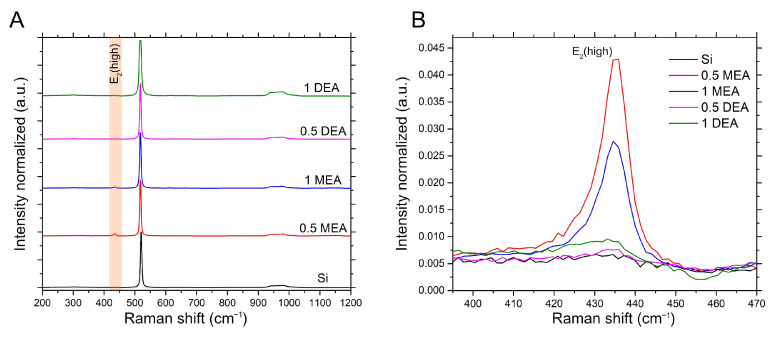
Raman spectra for samples grown on silicon. (**A**) The whole range of spectra normalized to the most intensive peak; (**B**) the spectrum section for the fragment marked in orange on A. showing a comparison of the relative intensity of the E${}_{2}$(high).

**Figure 10 gels-08-00512-f010:**
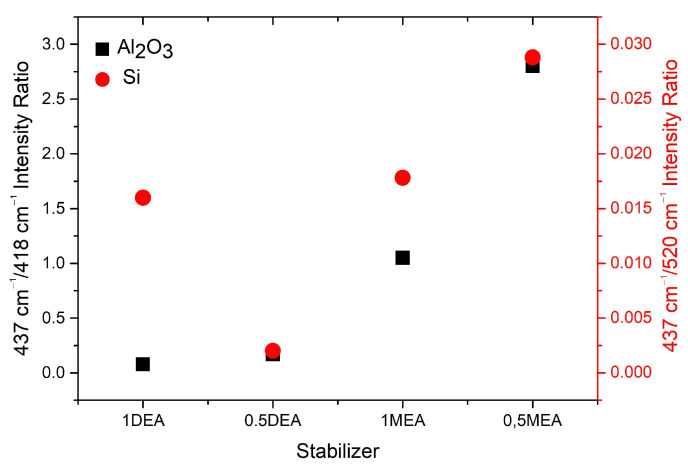
Ratio of intensities of ZnO E${}_{2}$(high) mode to the most prominent peak of substrate for sapphire (black) and silicon (red).

**Figure 11 gels-08-00512-f011:**
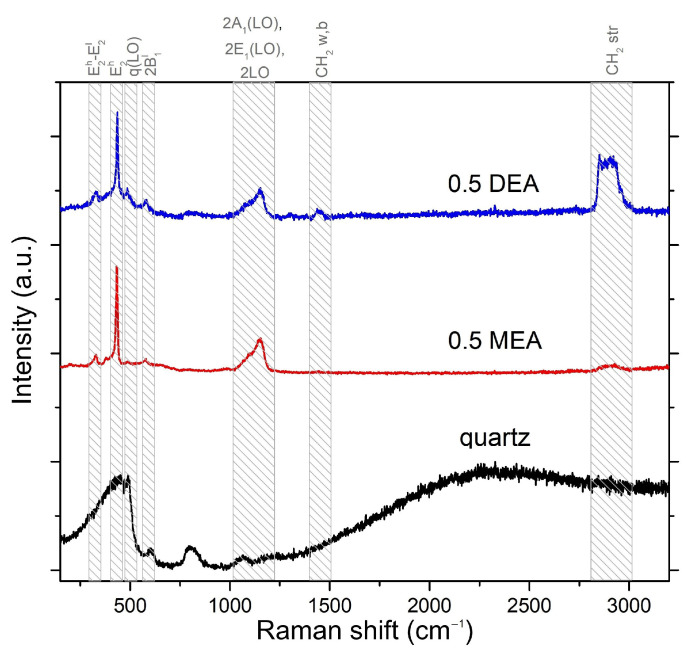
Raman spectra for samples grown on quartz. The grey tags relate to the ZnO and organic bands.

**Figure 12 gels-08-00512-f012:**
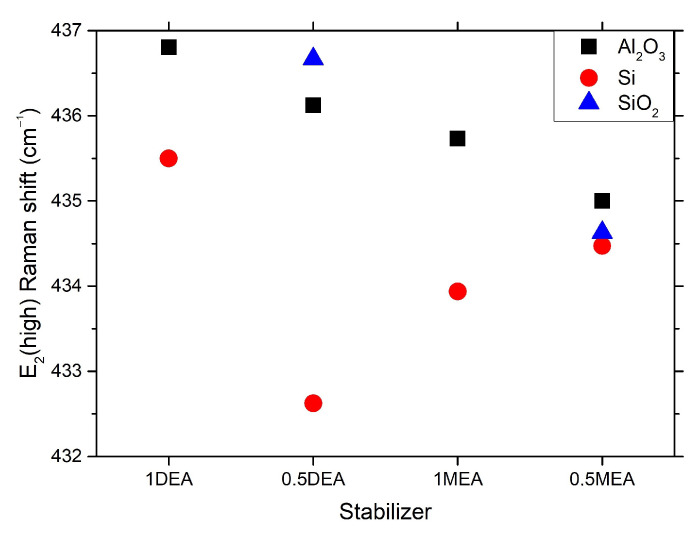
ZnO E${}_{2}$(high) mode Raman shift in regard to substrate, stabilizer, and its concentration.

**Figure 13 gels-08-00512-f013:**
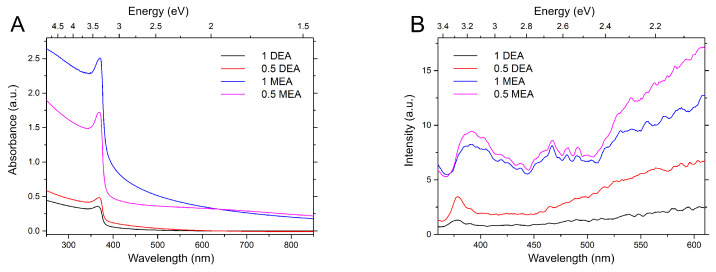
Absorbance (**A**) and photoluminescence (**B**) of samples grown on sapphire.

**Figure 14 gels-08-00512-f014:**
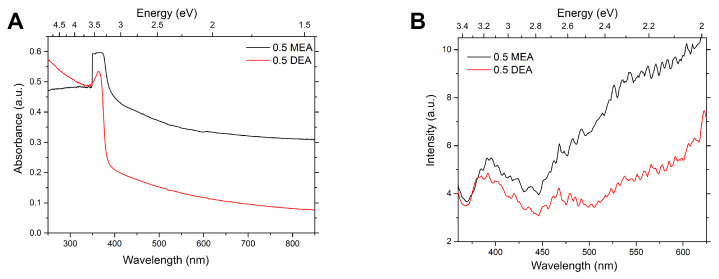
Absorbance (**A**) and photoluminescence (**B**) of samples grown on quartz.

**Figure 15 gels-08-00512-f015:**
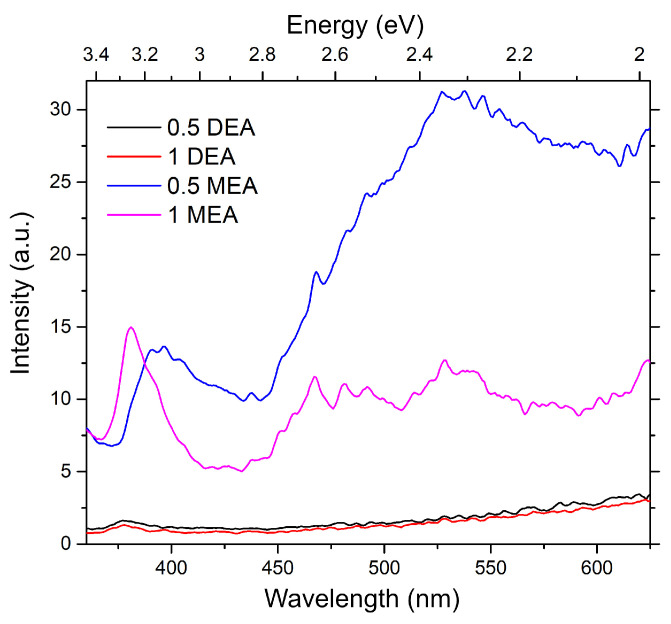
Photoluminescence of samples grown on silicon.

**Table 1 gels-08-00512-t001:** Mean crystallites lengths (in nm) and standard deviation estimated from SEM micrographs for ZnO layers grown on specific substrates.

	Al${}_{2}$O${}_{3}$	Si
0.5 MEA	90 ± 39 nm	119 ± 38 nm
1 MEA	121 ± 40 nm	82 ± 36 nm
0.5 DEA	116 ± 49 nm	74 ± 34 nm
1 DEA	91 ± 40 nm	72 ± 32 nm

**Table 2 gels-08-00512-t002:** The thickness of samples and their standard deviation estimated from confocal micrographs from ZnO layers grown on specific substrates. Thickness was measured in 10 places and averaged.

	Al${}_{2}$O${}_{3}$	Si	SiO${}_{2}$
0.5 MEA	18 ± 3 µm	20 ± 3 µm	15 ± 2 µm
1 MEA	17 ± 3 µm	20 ± 3 µm	
0.5 DEA	11 ± 3 µm	11 ± 3 µm	10 ± 3 µm
1 DEA	5 ± 3 µm	12 ± 3 µm	

**Table 3 gels-08-00512-t003:** The estimation of lattice parameters for ZnO layers grown on specific substrates.

Al${}_{2}$O${}_{3}$	a (Å)	c (Å)	c/a	Δ(c/a)
0.5 MEA	3.2455	5.1919	1.5997	0.0014
1 MEA	3.2471	5.1982	1.6009	0.0013
0.5 DEA	3.2354	5.1947	1.6056	0.0001
1 DEA	3.2452			
**Si**	**a (Å)**	**c (Å)**	**c/a**	**Δ(c/a)**
0.5 MEA	3.2436	5.1910	1.6004	0.0017
1 MEA	3.2432	5.1964	1.6022	0.0010
0.5 DEA	3.2454	5.3327	1.6431	0.0021
1 DEA	3.2447	5.3318	1.6432	0.0001

## Data Availability

The data may be supplemented via e-mail: ewelina.nowak@put.poznan.pl.

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
