# Peer review of "The Influence of Aminoalcohols on ZnO Films’ Structure"

_gels, 2022, doi:10.3390/gels8080512_

Round 1
Reviewer 1 Report
Nowak and her co-workers present in this work a study on the properties of ZnO layers grown using different aminoalcohols at different concentration rates, and on three different substrates (silicon, sapphire, and quartz glass), by spin-coating and annealing a zinc acetate solution. The structure of ZnO layers was estimated from morphology micrographs and Raman spectra. UV-vis absorption and photoluminescence spectroscopies were used to evaluate structural defects.
In general, the work is routine, experiments are simple, and the applied techniques for structural studies are limited; therefore, not surprisingly, the discussion and analysis in the manuscript is mainly descriptive and hypothetical. The application of powder X-ray diffraction, for crystalline phase analysis and identification, and high-resolution microscopy, preferably SEM, for obtaining information on the morphology, would have been essential. Thermogravimetric (TG) analysis of the dried precursor decomposition could also assist in understanding the degradation process and selecting the right temperature for annealing.
Further comments:
--- the selection of annealing temperature and time must be justified.
--- Figure 3 and 4: these figures are not informative at all; authors should use higher magnification.
--- Figure 6: ‘Figure 5’ in caption should be ‘Figure 6B’.
--- Figure 5, 6, 7, 8, 9, 10: the left parenthesis is missing, or the right parenthesis has to be replaced by a full stop at the end of the caption.
--- Figure 10: the capture is not in line with the figure; only the photoluminescence spectrum is shown.
--- line 297: ‘delegate’ is possibly not the right word in the context; please replace.
--- although the intensity of Raman bands of ZnO for samples prepared at the presence of MEA and DEA on quartz are comparable (Figure 7), this is not the case for samples prepared on sapphire and silicon substrates; in these latter cases, Raman bands of ZnO for samples prepared at the presence of DEA are negligible small (Figure 5 and 6); please explain.
--- A final reading of the manuscript is required to improve the English of the text and to remove typos.
Author Response
We are grateful for your time and constructive comments on our manuscript. We have implemented your suggestions and wish to submit a revised version of the manuscript for further consideration in the journal. Changes in the initial version of the manuscript are either highlighted for added sentences. Below, we also provide a point-by-point response explaining how we have addressed each of your comment:
Nowak and her co-workers present in this work a study on the properties of ZnO layers grown using different aminoalcohols at different concentration rates, and on three different substrates (silicon, sapphire, and quartz glass), by spin-coating and annealing a zinc acetate solution. The structure of ZnO layers was estimated from morphology micrographs and Raman spectra. UV-vis absorption and photoluminescence spectroscopies were used to evaluate structural defects.
In general, the work is routine, experiments are simple, and the applied techniques for structural studies are limited; therefore, not surprisingly, the discussion and analysis in the manuscript is mainly descriptive and hypothetical. The application of powder X-ray diffraction, for crystalline phase analysis and identification, and high-resolution microscopy, preferably SEM, for obtaining information on the morphology, would have been essential. Thermogravimetric (TG) analysis of the dried precursor decomposition could also assist in understanding the degradation process and selecting the right temperature for annealing.
Thank you very much for the time devoted to the review of the work. As suggested, additional tests - XRD and SEM - were performed to make the work more comparative.
Further comments:
--- the selection of annealing temperature and time must be justified.
The experiment based on comparing time and temperatures of annealing was conducted by us in different manuscript, which is currently under review. The 400 °C was the lowest round temperature, which enabled to achieve polycrystalline layer without organic contamination. We have also investigated (doi.org/10.3390/cryst11050533), that higher temperatures may influence the diffiusion of substrate ions into a lattice. Thus it was decided to use the lowest possible temperature to exclude their influence on the structure. The annealing time was the shortest time to to obtain the transition from amorphous to polycrystalline structure for DEA stabilized samples.
--- Figure 3 and 4: these figures are not informative at all; authors should use higher magnification.
The pictures were lightened in external software, which allowed for more accurate images. additionally, the measurements were supplemented with SEM with a magnification of 100,000x.
--- Figure 6: ‘Figure 5’ in caption should be ‘Figure 6B’.
Thank you very much! We have corrected the error.
--- Figure 5, 6, 7, 8, 9, 10: the left parenthesis is missing, or the right parenthesis has to be replaced by a full stop at the end of the caption.
Thank you very much for this remark – the correction was made
--- Figure 10: the capture is not in line with the figure; only the photoluminescence spectrum is shown.
Thank you very much for this remark – the correction was made
--- line 297: ‘delegate’ is possibly not the right word in the context; please replace
Thank you very much for this remark – the correction was made
--- although the intensity of Raman bands of ZnO for samples prepared at the presence of MEA and DEA on quartz are comparable (Figure 7), this is not the case for samples prepared on sapphire and silicon substrates; in these latter cases, Raman bands of ZnO for samples prepared at the presence of DEA are negligible small (Figure 5 and 6); please explain.
Thank you very much for this remark. Due to further analysis, we have noticed both a much smaller thickness of the DEA-based layer and the appearance of additional phases in it. Thus, the analysis of ZnO E2(high) mode relative intensity, as well as its frequency shifting, was additionally conducted.
--- A final reading of the manuscript is required to improve the English of the text and to remove typos.
Thank you, we have tried to improve the English of the text.
Reviewer 2 Report
The study investigates the effect of aminoalcohol additives on the structural properties of ZnO films prepared using sol-gel method. The authors used various optical methods to estimate the structural changes of the prepared films. They also presented an interesting discussion based on Raman spectroscopy results. However, the following comments must be addressed before the manuscript can be considered for publication:
1. From figure 2 the authors concluded that the use of low stabilizer concentration leads to film with larger grains. However, it is difficult to draw the same conclusion for DEA as the images in 2A & 2B are not clear.
2. The images in 3A & 3B are also not clear to give information about the effect of DEA concentration.
3. In figure 4, the morphological differences between the sample with MEA and that with DEA are not clear. Moreover, the effect of high MEA & DEA concentration (10%) on the samples deposited on quartz is not shown in the figure.
4. The samples should be imaged with a scanning electron microscope to estimate their morphology and to clearly show the structural changes induced by the use of the different additives/additive concentrations. Estimation of samples thickness is also recommended.
5. In figure 7, although both bands, E2 (high at 437 cm-1) and E2 (high)-E2 (low) at 330 cm-1, are present in the Raman spectra of the samples prepared with MEA and DEA, the authors stated that “indicates a much more significant defect in the layer produced with the use of
DEA”. The authors must explain this point. They should also present and discuss the data for these samples with high additives concentration (10%).
6. The authors should compare the variation in the intensity of the E2 (high) band for the samples with MEA & DEA prepared on the different substrates and discuss the role of the used substrate.
7. The authors should present and discuss the absorption and photoluminescence data for the samples with high additives concentration (10%) in figure 9.
8. Figure 10 presents only the photoluminescence of the samples. Correct the figure caption.
9. The samples should be characterized in details using XRD to support the discussion in the parts 3.2, 3.3 and 4 of the manuscript.
Author Response
We are grateful for your time and constructive comments on our manuscript. We have implemented your suggestions and wish to submit a revised version of the manuscript for further consideration in the journal. Changes in the initial version of the manuscript are either highlighted for added sentences. Below, we also provide a point-by-point response explaining how we have addressed each of your comment:
The study investigates the effect of aminoalcohol additives on the structural properties of ZnO films prepared using sol-gel method. The authors used various optical methods to estimate the structural changes of the prepared films. They also presented an interesting discussion based on Raman spectroscopy results. However, the following comments must be addressed before the manuscript can be considered for publication:
From figure 2 the authors concluded that the use of low stabilizer concentration leads to film with larger grains. However, it is difficult to draw the same conclusion for DEA as the images in 2A & 2B are not clear.
Thank you very much for this suggestion. For a better comparison, we have prepared the SEM micrographs.
The images in 3A & 3B are also not clear to give information about the effect of DEA concentration.
Thank you very much for the remark. As we wrote in a reply to Reviewer 1, the pictures were lightened in external software, which allowed for more accurate images. additionally, the measurements were supplemented with SEM with a magnification of 100,000x, which were analysed.
In figure 4, the morphological differences between the sample with MEA and that with DEA are not clear. Moreover, the effect of high MEA & DEA concentration (10%) on the samples deposited on quartz is not shown in the figure.
The effect of high concentration of MEA and DEA on samples deposited on quartz is not shown, because it wasn’t analysed in following article. Unfortunatelly, currently we are unable to produce additional samples due to the shortage on quartz substrate. Nevertheless, if the quartz samples disturb the continuity of the article, we can omit them in the analysis
The samples should be imaged with a scanning electron microscope to estimate their morphology and to clearly show the structural changes induced by the use of the different additives/additive concentrations. Estimation of samples thickness is also recommended.
Thank you very much for this suggestion. The SEM micrographs and estimation of layer thickness are included in a manuscript.
In figure 7, although both bands, E2 (high at 437 cm-1) and E2 (high)-E2 (low) at 330 cm-1, are present in the Raman spectra of the samples prepared with MEA and DEA, the authors stated that “indicates a much more significant defect in the layer produced with the use of DEA”. The authors must explain this point. They should also present and discuss the data for these samples with high additives concentration (10%).
The appearance of defects may be indicated by the appearance of the second harmonic of B1 mode, which is much more prominent in the case of the sample grown with DEA stabilizer, which was stated already in a previous version of the manuscript.
The authors should compare the variation in the intensity of the E2 (high) band for the samples with MEA & DEA prepared on the different substrates and discuss the role of the used substrate.
Thank you very much for this suggestion, the analysis is included in the manuscript.
The authors should present and discuss the absorption and photoluminescence data for the samples with high additives concentration (10%) in figure 9.
Thank you for this remark. We agree, that the discussion about samples with higher concentrations of stabilizers would enrich the article. However, as we stated, currently we are unable to produce additional samples due to the shortage on a quartz substrate.
Figure 10 presents only the photoluminescence of the samples. Correct the figure caption
Thank you for this remark. The correction is made.
The samples should be characterized in details using XRD to support the discussion in the parts 3.2, 3.3 and 4 of the manuscript.
Thank you very much for your suggestion. The XRD measurements are included in a new version the a manuscript.
Reviewer 3 Report
The manuscript “The Influence of Aminoalcohols on ZnO Films Structure” written by Ewelina Nowak et al. reports the preparation and characterization of ZnO layers grown using different amino alcohols at different concentrations rates. The paper is interesting, but it should be improved.
I have the following observations:
1. The authors should rewrite the abstract and introduction clearly express the paper's aim, novelty, and results
2. The authors should expand their study and the results in their own words, not through the use of references as in the discussion section.
3. X-ray diffraction and Scanning Electron Microscopy determinations should be included in the manuscript.
4. The conclusions should be supported by the results obtained.
5. The authors stated: “The change in the lattice parameters may be related primarily to the redshift of the absorption maximum ad the NBE band in the photoluminescence spectra for the samples produced based on MEA versus those made with DEA.” The lattice parameters are not presented in the manuscript.
Author Response
We are grateful for your time and constructive comments on our manuscript. We have implemented your suggestions and wish to submit a revised version of the manuscript for further consideration in the journal. Changes in the initial version of the manuscript are either highlighted for added sentences. Below, we also provide a point-by-point response explaining how we have addressed each of your comment:
The manuscript “The Influence of Aminoalcohols on ZnO Films Structure” written by Ewelina Nowak et al. reports the preparation and characterization of ZnO layers grown using different amino alcohols at different concentrations rates. The paper is interesting, but it should be improved.
I have the following observations:
- The authors should rewrite the abstract and introduction clearly express the paper's aim, novelty, and results
Thank you very much for this suggestion. We have improve the abstract and added a paragraph to Introduction which helps to express the aim of the article.
- The authors should expand their study and the results in their own words, not through the use of references as in the discussion section.
Thank you very much for this suggestion. We have added the analysis of our measurements to a discussion. However, we believe that comparing the results with the literature data is extremely important for understanding the issue and may be an advantage of the work.
- X-ray diffraction and Scanning Electron Microscopy determinations should be included in the manuscript.
Thank you very much for this suggestion. The SEM and XRD are included to a new version of the manuscript.
- The conclusions should be supported by the results obtained.
Thank you for a remark. The conclusions are supplemented with research conclusions.
- The authors stated: “The change in the lattice parameters may be related primarily to the redshift of the absorption maximum ad the NBE band in the photoluminescence spectra for the samples produced based on MEA versus those made with DEA.” The lattice parameters are not presented in the manuscript.
Thank you very much for the suggestion. The lattice parameters are calculated from XRD patterns and included to a manuscript.
Round 2
Reviewer 1 Report
Authors have corrected the manuscript according to reviewer's comments. The manuscript has been sufficiently improved. I suggest accepting the manuscript in its present form.
Reviewer 2 Report
The authors have addressed most of the comments of this reviewer.
Reviewer 3 Report
The authors have substantially and satisfactory revised the manuscript.